# The Diagnostic Accuracy of Artificial Intelligence in Radiological Markers of Normal-Pressure Hydrocephalus (NPH) on Non-Contrast CT Scans of the Brain

**DOI:** 10.3390/diagnostics13172840

**Published:** 2023-09-01

**Authors:** Dittapong Songsaeng, Poonsuta Nava-apisak, Jittsupa Wongsripuemtet, Siripra Kingchan, Phuriwat Angkoondittaphong, Phattaranan Phawaphutanon, Akara Supratak

**Affiliations:** 1Department of Radiology, Faculty of Medicine Siriraj Hospital, Mahidol University, Bangkok 10700, Thailand; dsongsaeng@gmail.com (D.S.);; 2Faculty of Information and Communication Technology, Mahidol University, Salaya, Nakhon Pathom 73170, Thailand

**Keywords:** NPH, radiologic markers, hydrocephalus, AI

## Abstract

Diagnosing normal-pressure hydrocephalus (NPH) via non-contrast computed tomography (CT) brain scans is presently a formidable task due to the lack of universally agreed-upon standards for radiographic parameter measurement. A variety of radiological parameters, such as Evans’ index, narrow sulci at high parietal convexity, Sylvian fissures’ dilation, focally enlarged sulci, and more, are currently measured by radiologists. This study aimed to enhance NPH diagnosis by comparing the accuracy, sensitivity, specificity, and predictive values of radiological parameters, as evaluated by radiologists and AI methods, utilizing cerebrospinal fluid volumetry. Results revealed a sensitivity of 77.14% for radiologists and 99.05% for AI, with specificities of 98.21% and 57.14%, respectively, in diagnosing NPH. Radiologists demonstrated NPV, PPV, and an accuracy of 82.09%, 97.59%, and 88.02%, while AI reported 98.46%, 68.42%, and 77.42%, respectively. ROC curves exhibited an area under the curve of 0.954 for radiologists and 0.784 for AI, signifying the diagnostic index for NPH. In conclusion, although radiologists exhibited superior sensitivity, specificity, and accuracy in diagnosing NPH, AI served as an effective initial screening mechanism for potential NPH cases, potentially easing the radiologists’ burden. Given the ongoing AI advancements, it is plausible that AI could eventually match or exceed radiologists’ diagnostic prowess in identifying hydrocephalus.

## 1. Introduction

Hydrocephalus is a condition where there is an abnormal accumulation of cerebrospinal fluid (CSF) within the brain’s ventricles, resulting in an increased intracranial pressure. It is classified into two types; obstructive hydrocephalus, which occurs when there is a physical blockage in the cerebrospinal fluid (CSF) flow pathway, and communicating hydrocephalus, which is characterized by abnormal CSF accumulation in the ventricles and subarachnoid spaces due to defects in reabsorption. Normal-pressure hydrocephalus (NPH) is a subtype of communicating hydrocephalus, which is typically seen in older adults and can present with symptoms such as gait disturbance, urinary incontinence, and memory impairment. These symptoms can be mistaken for Alzheimer’s disease or other types of dementia, making NPH a challenging diagnosis [1].

NPH is categorized into two types. The first is idiopathic NPH (iNPH), which is caused by an unknown reason that affects the reabsorption of cerebrospinal fluid back into the venous system. The second type is secondary NPH, which results from various factors such as bleeding in the brain’s cerebrospinal fluid, head trauma, infection, tumor, or complications of surgery. These factors can cause the accumulation of cerebrospinal fluid in the brain’s ventricles and subarachnoid spaces, leading to NPH symptoms such as gait disturbance, urinary incontinence, and memory impairment.

Several studies have investigated the use of radiological parameters from non-contrast computed tomography (NCCT) of the brain for the diagnosis of hydrocephalus. Nevertheless, diagnosing NPH based on radiographic imaging remains challenging due to the lack of standardized measurement methods for these radiographic parameters [2]. Various radiological parameters, which are used to diagnose NPH in our study, included Evans’ index, narrow sulci at high parietal convexity, the dilatation of the Sylvian fissures, focally enlarged sulci, the widening of temporal horns, callosal angle, and periventricular hypodensities [3].

With the increasing application of technology in the medical field, artificial intelligence (AI) has emerged as a promising tool for improving diagnostic accuracy and reducing errors in radiology. AI can quickly analyze large amounts of medical data, such as imaging studies, and identify abnormalities that might be missed by human radiologists. By utilizing deep learning algorithms, AI can recognize patterns and anomalies in medical images, providing more accurate and efficient diagnoses [4].

A typical pipeline for applying AI in NPH diagnosis involves three key steps. Firstly, it involves the identification of distinct regions, with a specific focus on the CSF and ventricular system in MRI [5] and CT [6] scans via medical imaging software or segmentation models. Secondly, the pipeline entails determining essential volumetric features extracted from brain regions, such as CSF and ventricles. Finally, machine learning algorithms are trained on the extracted features to establish relationships between NPH and non-NPH groups. Another potential biomarker for diagnosing NPH is the pattern of hypometabolism detected through positron emission tomography (PET) scans [7,8]. However, it has not been widely explored due to its invasive nature.

Siriraj Hospital of Mahidol University is utilizing AI innovations for screening pulmonary tuberculosis in chest radiographs and ASPECT in patients with clinically suspected acute cerebrovascular ischemia. Currently, the hospital is conducting research to compare the sensitivity, specificity, positive predictive value, negative predictive value, and accuracy of AI to radiological parameters measured by radiologists in normal and NPH-diagnosed groups. This study aims to investigate how the typical AI pipeline can improve NPH diagnosis, and gain valuable insights into the potential benefits and limitations of using AI.

## 2. Materials and Methods

### 2.1. Patient Selection

A retrospective study carried out between December 2012 and August 2022, which included all patients over the age of 18 who had both clinical data and imaging available. NPH was confirmed in patients via the gold standard method for diagnosis: cerebrospinal fluid (CSF) tap test. Patients with intracranial mass, cerebral hemorrhage, and large cerebral infarction leading to anatomical distortion of the brain were excluded from the study.

### 2.2. Imaging Review

CT imaging was assessed for all patients at the time of initial diagnosis. The images were reviewed by two senior neuroradiologists with more than 20 and 10 years of working experience and a third-year radiology resident, who were blinded to the patients’ clinical status, using the department’s Picture Archiving and Communication System (PACS). Inter-observer agreement was evaluated between the two neuroradiologists and the resident. In cases of disagreement, the final judgement was made by consensus. The AI was also used to evaluate the same groups of patients and identify cases of NPH.

Seven radiological parameters were used in this study, including Evans’ index, narrow sulci at high parietal convexity, dilatation of the Sylvian fissures, focally enlarged sulci, widening of temporal horns, callosal angle, and periventricular hypodensities. Each radiological parameter was separately converted into a point system with cut-off values based on earlier studies [3] and total scores were calculated, ranging from 0 to 12 points. The study also compared the reliability of each imaging feature alone with that of the overall iNPH Radscale score. To standardize measurements of each radiologic parameters, the planes were carefully aligned with anatomical landmarks in both axial and coronal planes. The axial plane was positioned parallel to the pituitary–fastigium (of the fourth ventricle) axis, while the coronal plane was angulated perpendicular to the transverse plane for all measurements except for the callosal angle, which required a coronal plane perpendicular to the intercommissural plane [3].

### 2.3. Radiologic Parameters

The following radiological parameters were evaluated by two radiologists and a third-year radiology resident (Figure 1 for atlas of measurements and scoring levels [3]).

Evans’ index: The ratio between the maximal width of the frontal horns of the lateral ventricles (B–C) by the maximal width of the inner table of the cranium in the same axial image [9].Narrow parietal sulci: At high-convexity and parafalcine region assessed in both axial planes in the most superior slices and coronal plane [10].Dilation of the Sylvian fissures: Reported as present or not present in the coronal plane compared with surrounding sulci [11].Focally enlarged sulci: Compared with surrounding sulci, usually found in coronal or axial planes [12].Temporal horns: Reported as mean width of the right and left side, measuring in the axial plane [11].Callosal angle: Angle between the lateral ventricles in the coronal plane through the posterior commissure perpendicular to the intercommissural plane [13].Periventricular hypodensities: Along the lateral ventricles graded as not present, present as a cap around frontal horns or confluently extending around the lateral ventricles [14].

### 2.4. AI Evaluation

Our AI method comprises three main steps. Firstly, we trained a modified 2D U-Net model [15] for CSF segmentation using a noisy dataset generated from a medical imaging software, named SPM12 [16]. The use of such noisy dataset is to facilitate the model training without asking experienced radiologists to annotate the CT scans. The use of such weakly supervised segmentation model demonstrated a better CSF segmentation performance in our initial experiment than directly using the outputs from SPM12. Secondly, the outputs from the segmentation model were used to extract volumetric features. Finally, the extracted features were used to train a NPH classification model.

The modified 2D U-Net model consists of four encoder blocks, a bottleneck, and four decoder blocks. It was trained using the sigmoid focal cross-entropy loss function [17] and was initialized with weights using He normalization [18]. The Adam optimizer [19] with an initial learning rate of 0.001 was used to train the model for up to 200 epochs with an early stopping based on the performance on the validation set.

The extracted features were classified into two categories, namely global features representing entire brain region characteristics and local features representing entire brain region characteristics. The study aimed to analyze the impact of both global whole-brain volume metrics and local partition-brain metrics on NPH classification. 

The parameters for each feature in cerebrospinal fluid (CSF), White and Grey (WG) ratio, and standard deviation (Std) are shown below:

CSF ratio all = $\frac{N_{\mathrm{CSF}}}{N_{\mathrm{CSF}}+{\;N}_{\mathrm{white}\&\mathrm{grey}}}$

CSF/WG ratio = $\frac{N_{\mathrm{CSF}}}{N_{\mathrm{white}\&\mathrm{grey}}}$

CSF size = $\frac{N_{\mathrm{CSF}}}{\mathrm{image}\_\mathrm{size}}$

Brain size = $\frac{N_{\mathrm{white}\&\mathrm{grey}}}{\mathrm{image}\_\mathrm{size}}$

Mean CSF ratio = $\frac{\sum_{i=0}^{n}\mathrm{Xi}}{n}$; $\mathrm{Xi}\;=\frac{N_{\mathrm{CSF}}}{N_{\mathrm{CSF}}+{\;N}_{\mathrm{white}\&\mathrm{grey}}}$

Min CSF ratio = Min of $\frac{N_{\mathrm{CSF}}}{N_{\mathrm{CSF}}+{\;N}_{\mathrm{white}\&\mathrm{grey}}}$

Max CSF ratio = Max of $\frac{N_{\mathrm{CSF}}}{N_{\mathrm{CSF}}+{\;N}_{\mathrm{white}\&\mathrm{grey}}}$

Std CSF ratio = $\begin{array}{c} \sqrt{\frac{\sum {\left|x-\;\bar{x}\right|}^{2}}{n}} \end{array}$; $x=\frac{N_{\mathrm{CSF}}}{N_{\mathrm{CSF}}+{\;N}_{\mathrm{white}\&\mathrm{grey}}}$

Mean CSF/WG ratio = $\frac{\sum_{i=0}^{n}\mathrm{Xi}}{n}$; $\mathrm{Xi}\;=\frac{N_{\mathrm{CSF}}}{{\;N}_{\mathrm{white}\&\mathrm{grey}}}$

Min CSF/WG ratio = Min of $\frac{N_{\mathrm{CSF}}}{N_{\mathrm{white}\&\mathrm{grey}}}$

Max CSF/WG ratio = Max of $\frac{N_{\mathrm{CSF}}}{N_{\mathrm{white}\&\mathrm{grey}}}$

Std CSF/WG ratio = $\begin{array}{c} \sqrt{\frac{\sum {\left|x-\;\bar{x}\right|}^{2}}{n}} \end{array}$; $x=\frac{N_{\mathrm{CSF}}}{N_{\mathrm{white}\&\mathrm{grey}}}$

This evaluation utilizes $N_{\mathrm{CSF}}$ and $N_{\mathrm{white}\&\mathrm{grey}}$ to represent the number of CSF and white/gray matter pixels within the segmentation masks. Meanwhile, image_size indicates the overall number of pixels in a brain slice.

The extracted global and local volumetric features were used to train an NPH classification model. In particular, a logistic regression model was trained to perform a binary classification (1 = NPH and 0 = non-NPH) using stratified 5-fold cross-validation on 227 NPH and 110 normal data. The model was used a regularization parameter c = 10, and feature selection was performed using the chi-squared (chi2) method, resulting in the selection of 10 features: ‘CSF ratio_5’, ‘CSF ratio_4’, ‘CSF ratio_6’, ‘CSF ratio_9’, ‘Std CSF ratio_9’, ‘CSF ratio_3’, ‘CSF ratio_2’, ‘CSF ratio_1’, ‘CSF ratio_8’, ‘CSF_ratio_all’. The partitions 0–9 indicate different levels of the brain, with 0 being the lowest (closest to the neck) and 9 being the highest (at the top of the head).

To gain insights into the features that predominantly influenced our AI model’s predictions, we utilized the SHAP library [https://github.com/slundberg/shap accessed on 12 April 2023]. This allowed us to assess the impact of each feature when predicting the probability of NPH for each CT scan. Our analysis revealed that the three most influential features were CSF ratio_8, CSF ratio_5, and CSF ratio_4 (refer to Figure 2). CSF ratio_8 captures changes in focal sulcal enlargement, while CSF ratio_5 and CSF ratio_4 correspond to enlarged ventricular regions (see Figure 3). This finding indicates that our AI model focuses on the key areas commonly examined by neuroradiologists during NPH diagnosis, underscoring its alignment with expert practices.

### 2.5. Statistical Analysis

Clinical data and radiological findings are presented with descriptive statistics. Categorical data are present as numbers and percentages and compared using Pearson’s chi-squared test. Continuous data are reported as mean ± standard deviation (SD) and compared using independent *t*-test. A *p*-value of <0.05 was considered statistically significant. Both the normal and NPH groups were compared to assess the sensitivity, specificity, accuracy, PPV, NPV, and area under the receiving operating characteristic (ROC) curve between radiologists and AI. A binary logistic regression was used to determine the cut-off value for predicting whether the patient was normal, borderline, or had NPH.

## 3. Results

This study retrospectively enrolled 217 subjects, including 112 patients clinically confirmed with NPH who underwent the gold standard CSF closing pressure-guided tap test, and 105 normal patients. Among the NPH group, 108 patients were classified as iNPH, while only four patients are secondary NPH. The median age at the time of the clinical diagnosis was 76 years (range, 68–84 years); and 60 (57.1%) were men and 45 (42.9%) were women (Table 1). Clinical symptoms, including gait disturbance, urinary incontinence, and memory impairment, are statistically significant (*p* < 0.001) in the NPH group (Table 1). Univariate and multivariate analyses found that four radiological parameters (Evans’ index, dilated Sylvian fissures, focally enlarged sulci, widening temporal horns) and percentage of the total scores of radiologic parameter between normal and NPH groups were significantly associated with clinical symptoms with *p*-value < 0.0001 (Table 2 and Table 3). Binary logistic regression analysis indicated that total scores of <3 points, 3–4 points, and ≥5 points were likely to be considered normal, borderline, or patients with NPH (Table 4).

The sensitivity for radiologists and AI was 77.14% and 99.05%, respectively, with a specificity of 98.21% and 57.14%, respectively, under the cut-off value of 5. NPV, PPV, and accuracy for radiologists were 82.09%, 97.59%, and 88.02%, respectively, while for AI, these values were 98.46%, 68.42%, 77.42%, respectively (Table 5). The receiver operating characteristic (ROC) curve of the diagnostic index of radiological parameters, measured by radiologists for the diagnosis of NPH (Figure 4), demonstrated an area under the curve of 0.954 (*p* < 0.001), and the ROC AI (Figure 5) was 0.784 (*p* < 0.001). The narrow sulci of high parietal convexity, callosal angle, and periventricular hypodensities were omitted due to collinearity.

The odds ratios (ORs) for Evans’ index, dilated sylvian fissures, focally enlarged sulci, and widening temporal horns were found to be statistically significant. Multivariate analysis revealed ORs of 3.49 (1.07–11.42) and 38.37 (6.04–243.56) for Evans’ indexes of 1 and 2, respectively. The OR for dilated Sylvian fissures was 3.07 (1.04–9.08), while for focally enlarged sulci was 7.88 (1.28–48.25), for widening temporal horn was 5.35 (1.88–15.16), and 12.55 (2.15–73.31) for the first and second grades, respectively (Table 2).

## 4. Discussion

The study findings suggest that NPH is increasingly being diagnosed in elderly patients undergoing brain imaging for other reasons. This is consistent with recent epidemiological surveys in Sweden that reported a 3.7% prevalence of iNPH among individuals aged 65 years old [20]. NPH is more likely to occur in the elderly and is often associated with other age-related diseases such as hypertension, T2DM, Parkinson’s disease, and dementia [21]. The clinical symptoms showed that gait disturbance, urinary incontinence, and memory impairment are statistically significant (*p* < 0.001) in the NPH group. It should be noted that our study mainly focused on iNPH cases due to the small number of secondary NPH. 

Although various radiological parameters are used to help confirm the diagnosis, the cut-off values of each parameter are not well established. The morphologic features indicating the likelihood of morphologic features of NPH remain undefined. Furthermore, NPH commonly affects the elderly, who may have associated age-related brain atrophy. Therefore, the differential diagnosis between iNPH and cortical brain atrophy or small vessel disease is difficult [22]. Our study investigated the radiologic parameters in both NPH and normal groups of patients and compared the results found in AI. 

There is no united agreement in standardized measurement for each radiologic parameter. For example, there are no specific images used in the measurements of Evans’ index, which is the most widely used parameter in ventricular width. Because CT of the brain provides numerous axial images, the maximal width of the frontal horns can be measured on the same or different images by each radiologist [23]. Moreover, some radiologists measure the maximal width of the frontal horns and maximal inner diameter on the same axial image [24], while some measure maximal width in the separate planes [25]. Temporal width in our study is assessed in the axial plane and reported as mean width of the right and left side [11]. The measurement of its values may differ depending on the selection of images by each radiologist. Some of the radiologic parameters such as narrow sulci at high parietal convexity, the dilatation of the Sylvian fissures, focally enlarged sulci, and periventricular hypodensities are evaluated by using subjective methods, namely the visual rating score [26]. Dilated Sylvian fissures and focally enlarged sulci are frequently misinterpreted for cerebral atrophy [12]. However, the interobserver analysis was consistent in our study. The assessment of the callosal angle should ideally be measured on a coronal image perpendicular to anterior commissure–posterior commissure (AC-PC) plane at the level of the posterior commissure. However, minor differences in angular malrotations of the true coronal plane could affect the accurate measurement of the callosal angle [27].

Although periventricular hypodensities are a supporting feature of NPH, it is difficult to separate between white matter ischemia, which is commonly found in elderly patients with small vascular disease and subependymal effusion resulting from NPH, and results in the exclusion of patients from further NPH evaluation [28]. It can be seen that radiological parameters performed via different methods may have the different results [29]. Nowadays, the volumetric segmentation of CT brain scan methods are considered more accurate and are increasingly used in many studies [30,31].

The use of our AI method presents a new paradigm compared to the existing NPH diagnosis methods. Bianco et al. [5] employed Freesurfer software to extract volumetric features from MRI scans. This study, on the other hand, focuses on a more cost-effective imaging option: CT scan. Several recent studies have started to look at the potential use of CT scanning for building automated NPH classification models. Zhang et al. propose an automated method of predicting NPH using the volumetric segmentation of CT brain scans, which is the first method that automatically predicts NPH from CT scans using AI. The connectome data to compute features, which capture the impact of enlarged ventricles and regions of interest segmented from CT scans using AI, provide the fast and accurate volumetric segmentation of CT brain scans, which can thus improve the NPH diagnosis accuracy [32]. However, their approach relies on a 3D U-Net model for segmentation, which is more computationally expensive compared to our study. Additionally, their segmentation process necessitates training with manually annotated brain CT scans by radiologists. In contrast, our AI method relies on a noisy dataset generated from existing medical imaging tools, requiring zero annotation effort. In another study by Duan et al. [6], they developed a model for diagnosing hydrocephalus that incorporates clinical features such as Evan’s index and age along with CT images. Although their model demonstrated promising performance, the reliance on Evan’s index determined by radiologists could be a drawback. In contrast, our approach is more appealing, as it solely relies on CT images and is independent of radiologists during the segmentation model training and NPH prediction process.

Our study found that radiologists had a better diagnostic specificity, PPV, and overall accuracy than AI. However, AI volumetric segmentation demonstrated higher sensitivity in detecting ventricular enlargement, indicating its potential as a screening tool. Moreover, the accuracy of AI can be improved through a learning process that involves measuring brain volumes in an increasing number of patients. As AI technology continues to advance, it may become a valuable tool for diagnosing and managing NPH.

## 5. Limitation

As our study is a retrospective review of 10 years’ worth of data, there is a risk of chronological bias. Furthermore, certain radiological parameters such as narrow sulci at the high parietal convexity, callosal angle, and periventricular hypodensities were omitted due to collinearity. Therefore, a large-scale prospective study is needed to further investigate these parameters and confirm our findings.

Although our AI model did not perform as well as the consensus from three radiologists, it showed promise in the screening process by exhibiting higher sensitivity (recall). This potential use case could lead to a reduction in the number of patients requiring confirmation by radiologists.

Another aspect that this study has yet to explore is multicollinearity, which could impact the model’s interpretability, as indicated by the SHAP value. In our future research, we aim to address this issue by investigating more advanced models, like deep learning [33], that are less susceptible to multicollinearity.

## 6. Conclusions

Our study found that radiologists exhibited higher diagnostic sensitivity, specificity, PPV, and accuracy than AI. However, AI can serve as a screening tool in patients suspected of having NPH, reducing the workload on radiologists. Furthermore, AI’s accuracy can be enhanced through machine learning on an increasing number of brain volumetric measurements. In the future, AI may attain capabilities that are equivalent to or surpass those of radiologists in diagnosing hydrocephalus.

## Figures and Tables

**Figure 1 diagnostics-13-02840-f001:**
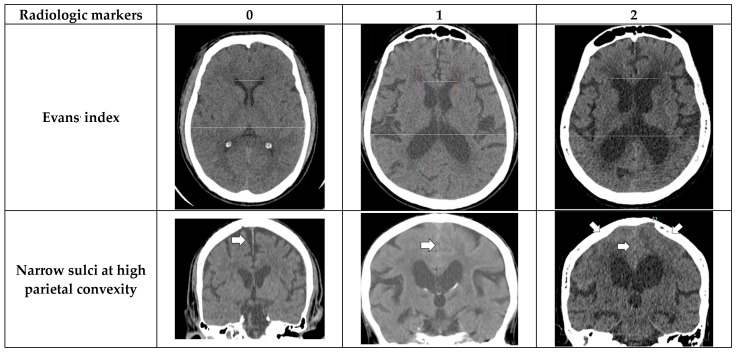
Imaging atlas with scoring according to the NPH scale [3].

**Figure 2 diagnostics-13-02840-f002:**
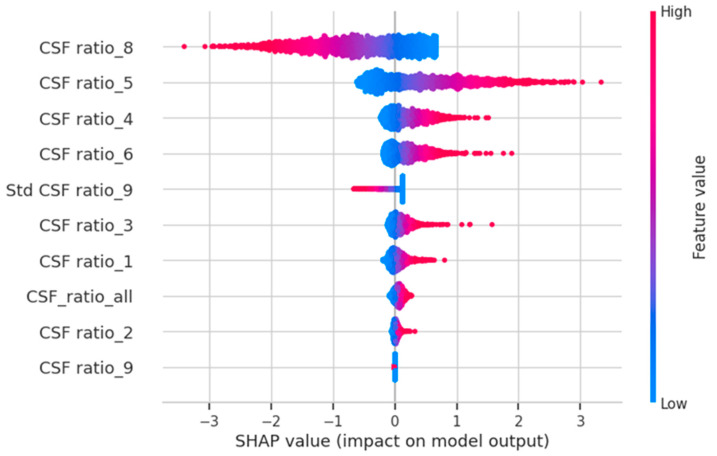
The SHAP values suggest that our AI model predictions were significantly influenced by the CSF ratio_8, CSF ratio_5, and CSF ratio_4, which are regions that neuroradiologists commonly focus on during the diagnosis process.

**Figure 3 diagnostics-13-02840-f003:**
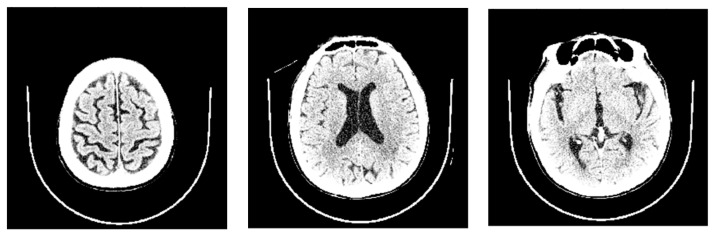
Samples of brain slice images from partitions 8, 5, and 4, arranged from left to right that visually demonstrate the specific brain regions associated with the CSF ratios that play a crucial role in NPH prediction.

**Figure 4 diagnostics-13-02840-f004:**
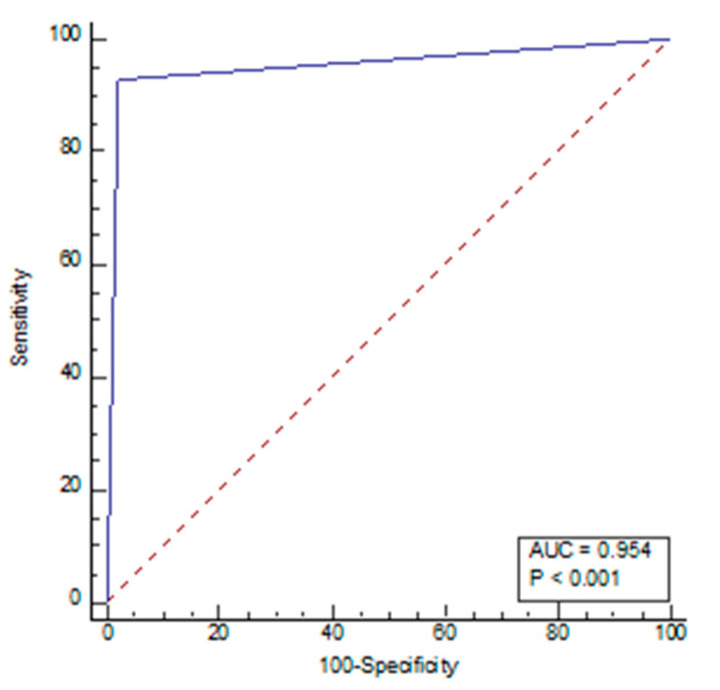
Receiver operating characteristic (ROC) curve of radiological parameters by radiologists.

**Figure 5 diagnostics-13-02840-f005:**
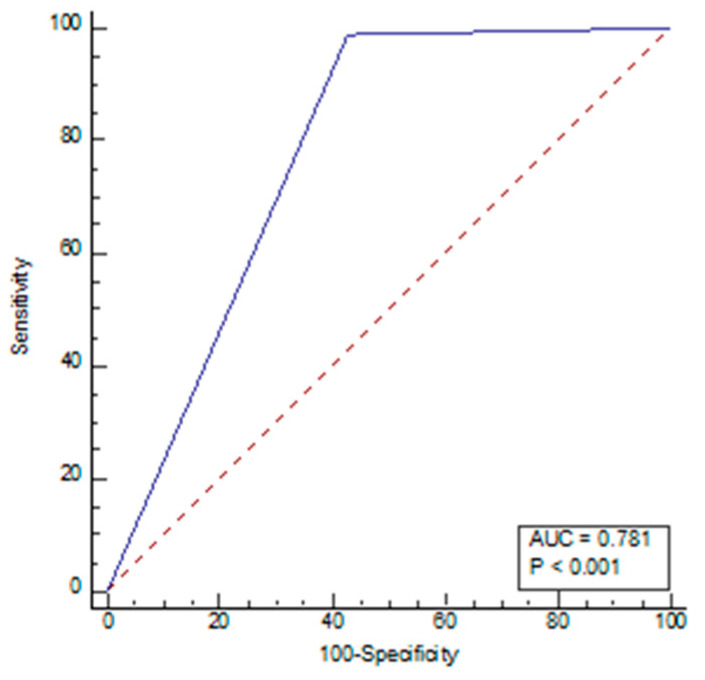
Receiver operating characteristic (ROC) curve of AI.

**Table 1 diagnostics-13-02840-t001:** Basic characteristics and demographic data between groups of normal and patients with NPH.

Variables	All (n = 217)	Normal (n = 112)	NPH (n = 105)	*p*-Value
Gender (M:F)	105 (48.4%):112 (51.6%)	55 (49.1%):57 (50.9%)	60 (57.1%):45 (42.9%)	0.236
Age (years)	65.4 ± 17.8	55.7 ± 19.2	75.7 ± 8.0	<0.001
Gait disturbance	99 (45.6%)	0 (0%)	99 (94.3%)	<0.001
Urinary incontinence	77 (35.5%)	0 (0%)	77 (73.3%)	<0.001
Memory impairment	61 (28.1%)	0 (0%)	61 (58.1%)	<0.001
HT *	122 (56.2%)	49 (43.8%)	73 (69.5%)	<0.001
T2DM	72 (33.2%)	26 (23.2%)	46 (43.8%)	<0.001
DLP	80 (36.9%)	42 (37.5%)	38 (36.2%)	0.842
Old CVA	42 (19.4%)	21 (18.8%)	21 (20.0%)	0.816
CKD	21 (9.7%)	1 (0.9%)	10 (9.5%)	0.941
CAD	20 (9.2%)	8 (7.1%)	12 (11.4%)	0.275
Parkinson’s disease	23 (10.6%)	0 (0%)	23 (21.9%)	<0.001
Dementia	20 (9.2%)	3 (2.7%)	17 (16.2%)	<0.001
OA knee	11 (5.1%)	6 (5.4%)	5 (4.8%)	0.842

* HT, hypertension; T2DM, type 2 diabetes mellitus; DLP, dyslipidemia; Old CVA, old cerebrovascular accident; CKD, chronic kidney disease; CAD, coronary artery disease; OA knee, osteoarthritis of the knee.

**Table 2 diagnostics-13-02840-t002:** Relationship of radiologic parameters to predict the likelihood of NPH.

Variable	^1^ Crude OR * (95% CI) **	*p*-Value	^2^ Adjusted OR (95% CI)	*p*-Value
Evans’ index		<0.0001		<0.0001
0	Ref. ***		Ref.	
1	12.77 (4.68–34.88)		3.49 (1.07–11.42)	
2	395.3 (73.91–2114.10)		38.37 (6.04–243.56)	
Dilatation of Sylvian fissures		<0.0001		<0.0001
0	Ref.		Ref.	
1	23.25 (11.12–48.62)		3.07 (1.04–9.08)	
Focally enlarged sulci		<0.0001		<0.0001
0	Ref.		Ref.	
1	25.499 (0.762–85.30)		7.88 (1.28–48.25)	
Widening temporal horns		<0.0001		<0.0001
0	Ref.		Ref.	
1	30 (12.83–70.13)		5.35 (1.88–15.16)	
2	132 (28.86–603.79)		12.55 (2.15–73.31)	

* OR, odds ratio; ** CI, confidence interval; *** Ref, reference. ^1^ Univariate analysis by Pearson’s chi-squared. ^2^ Multivariate analysis.

**Table 3 diagnostics-13-02840-t003:** Percentage of the total scores of radiologic parameters between normal and NPH groups.

Total Score	Normal	NPH	*p*-Value
0	46 (100%)	0	<0.0001
1	30 (96.8%)	1 (3.2%)	<0.0001
2	15 (75%)	5 (25%)	0.028
3	12 (63.2%)	7 (36.8%)	0.292
4	7 (38.9%)	11 (61.1%)	0.259
5	1 (5.6%)	17 (94.4%)	<0.0001
6	1 (5%)	19 (95%)	<0.0001
7	0	19 (100%)	<0.0001
8	0	11 (100%)	<0.0001
9	0	9 (100%)	0.002
10	0	4 (100%)	0.037
11	0	2 (100%)	0.142
12	0	0	N/A

**Table 4 diagnostics-13-02840-t004:** Scoring levels used for NPH prediction.

Score	Result of Predicted NPH
0–2	Negative
3–4	Borderline
≥5	Positive

**Table 5 diagnostics-13-02840-t005:** Comparison of sensitivity, specificity, NPV, PPV, and accuracy between radiologists and AI using score ≥ 5 as the cut-off value.

Variables	Radiologists	AI ***
Sensitivity	77.14%	99.05%
Specificity	98.21%	57.14%
NPV *	82.09%	98.46%
PPV **	97.59%	68.42%
Accuracy	88.02%	77.42%

* NPV, negative predictive value. ** PPV, positive predictive value. *** AI, artificial intelligence.

## Data Availability

Not applicable.

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
