# Peer review of "The Diagnostic Accuracy of Artificial Intelligence in Radiological Markers of Normal-Pressure Hydrocephalus (NPH) on Non-Contrast CT Scans of the Brain"

_diagnostics, 2023, doi:10.3390/diagnostics13172840_

Round 1
Reviewer 1 Report
This paper proposes an application of artificial intelligence on CT scan brain of NPH patients for improvement the diagnosis. There are several points need to be addressed to improve the quality of the manuscript.
Introduction: Breakdown of the existing techniques and literature gap
Authors did not mention about the overall existing neuroimaging techniques for NPH diagnosis. They did not provide a short breakdown of these techniques in AI with supporting references. Please provide this background in the introduction with the respective literature below to give a broader overview for the audience.
Use of Artificial Intelligence for NPH diagnosis:
[1] MR: 10.1016/j.parkreldis.2022.08.007
[2] PET: 10.3389/fneur.2021.700269
[3] CT: 10.1097/MD.0000000000021229
Materials and Methods:
AI evaluation
-Authors use global and local features, how they avoid multicollinearity? Which techniques they used?
-According to Figure 2, why did they consider only three most important features? Why not 4, 5?
-Since the dataset have a good size, why authors did not perform a split in train-test? Cross-validation should be used in the training model and validation in another balanced cohort.
Statistical Analysis
-Did authors check normal distribution?
Results
-line 190-193 : please provide more information on the collinearity.
-table 1: please better format the table since it it not clear that the t-test was performed between normal and NPH. I suggest to delete the first column.
Minor questions:
-Figure 4 and Figure 5.
According to table 5, the notes of each figure are inverted.
-line 237 , the number 3 is a reference? Please, check.
Author Response
Dear editor,
We have been through suggestions and questions by the reviewers and re-write part of the manuscript. In the part of the statistical analysis has been completed successfully with the Kolmogorov-Smirnov test for normality distribution. Hopefully, this is alright with you.
Thank you for your patience and kindness waiting for us.
Best regards,
Dittapong Songsaeng, MD.
Reviewer 2 Report
The manuscript is well written and this study emphasizes the valuable contribution that AI can make as an initial screening tool. Authors may consider including the discussion on ''By combining the expertise of radiologists with the capabilities of AI, there is an opportunity to improve the efficiency of diagnosis, alleviate the workload on radiologists, and ultimately enhance patient outcomes.''
Minor editing of English language required
Author Response
Dear editor,
We have been through suggestions and questions by the reviewers and re-write part of the manuscript. Hopefully, this is alright with you.
Thank you for your patience and kindness waiting for us.
Best regards,
Dittapong Songsaeng, MD.
Reviewer 3 Report
Well done for the great work. The topic and the results are novel and interesting.
Author Response

(The authors gave the same response as above.)

Reviewer 4 Report
1. The AI model used for the proposed diagnosis of NPH is not explained clearly.
2. There are n-number of AI models available for medical image classification. In this paper the general term AI model is used but not explained about the method.
3. The drawback in the AI model already available is not discussed or is it the first time tried for this specific problem?
4. If radiologists are good at doing the work, then what is the need for AI?
5. The conclusion is not impressive and does not give any information for the future researchers to work on.
6. The discussion session should bring out the analysis and insights/challenges of the work carried out.
7. Most of the references is quoted in the discussion session, which should be avoided.
8. The tabulated results are not validated with any mathematical modelling.
moderate English editing is required
Author Response

(The authors gave the same response as above.)

Round 2
Reviewer 1 Report
The revisions process has been successfully addressed by the authors.
Reviewer 4 Report
Corrections are fine. Accept.